# Tri-Herbal Medicine Divya Sarva-Kalp-Kwath (Livogrit) Regulates Fatty Acid-Induced Steatosis in Human HepG2 Cells through Inhibition of Intracellular Triglycerides and Extracellular Glycerol Levels

**DOI:** 10.3390/molecules25204849

**Published:** 2020-10-21

**Authors:** Acharya Balkrishna, Vivek Gohel, Rani Singh, Monali Joshi, Yash Varshney, Jyotish Srivastava, Kunal Bhattacharya, Anurag Varshney

**Affiliations:** 1Drug Discovery and Development Division, Patanjali Research Institute, Governed by Patanjali Research Foundation Trust, NH-58, Haridwar 249 405, Uttarakhand, India; pyp@divyayoga.com (A.B.); vivek.gohel@prft.co.in (V.G.); rani.singh@prft.co.in (R.S.); monali.joshi@prft.co.in (M.J.); yash.varshney@prft.co.in (Y.V.); jyotish.srivastava@prft.in (J.S.); kunalbhattacharya@prft.co.in (K.B.); 2Department of Allied and Applied Sciences, University of Patanjali, Patanjali Yog Peeth, Roorkee-Haridwar Road, Haridwar 249 405, Uttarakhand, India; 3Patanjali Yog Peeth (UK) Trust, 40 Lambhill Street, Kinning Park, Glasgow G41 1AU, UK

**Keywords:** steatosis, hepatocytes, Ayurveda, Phyto-medicines, Sarva-Kalp-Kwath, Livogrit, HepG2 cells

## Abstract

Steatosis is characterized by excessive triglycerides accumulation in liver cells. Recently, application of herbal formulations has gained importance in treating complex diseases. Therefore, this study explores the efficacy of tri-herbal medicine Divya Sarva-Kalp-Kwath (SKK; brand name, Livogrit) in treating free fatty acid (FFA)-induced steatosis in human liver (HepG2) cells and rat primary hepatocytes. Previously, we demonstrated that cytosafe SKK ameliorated CCl_4_-induced hepatotoxicity. In this study, we evaluated the role of SKK in reducing FFA-induced cell-death, and steatosis in HepG2 through analysis of cell viability, intracellular lipid and triglyceride accumulation, extracellular free glycerol levels, and mRNA expression changes. Plant metabolic components fingerprinting in SKK was performed via High Performance Thin Layer Chromatography (HPTLC). Treatment with SKK significantly reduced the loss of cell viability induced by 2 mM-FFA in a dose-dependent manner. SKK also reduced intracellular lipid, triglyceride accumulation, secreted AST levels, and increased extracellular free glycerol presence in the FFA-exposed cells. SKK normalized the FFA-stimulated overexpression of SREBP1c, FAS, C/EBPα, and CPT1A genes associated with the induction of steatosis. In addition, treatment of rat primary hepatocytes with FFA and SKK concurrently, reduced intracellular lipid accumulation. Thus, SKK showed efficacy in reducing intracellular triglyceride accumulation and increasing extracellular glycerol release, along with downregulation of related key genetic factors for FFA-associated steatosis.

## 1. Introduction

Liver ailments are evolving rapidly due to modern sedentary lifestyle, food-habits, and dependence on medications for minor illnesses. Liver steatosis is prevalent in the majority of obese individuals and the most common pathological condition found in 33% of the adult population in the United States of America [1]. Development of steatosis often originates from high-fat diet intake and metabolic disorders leading to excessive deposition of lipids and triglycerides (TGs) in the hepatocytes of patients [2,3]. Free fatty acids (FFAs) in the diet also increase levels of ceramides, which are known to be involved in several pathways linked to inflammation, apoptosis, insulin resistance, oxidative stress, and progression of steatosis [4]. Prolonged persistence of steatosis may lead to the development of hepatic fibrosis and other non-hepatic (cardiovascular, cancer, and neurological) health complications. While the global prevalence of non-alcoholic fatty liver disease might be near to a billion cases, the prevailing variety in the pathological manifestation of steatosis disease leads it to be rather under-reported [5]. Palmitic acid and monounsaturated oleic acid represents the two most abundant FFA present in high-fat diets [6]. Intake of high levels of oleic acid and palmitic acids from food sources has been shown to induce steatosis in hepatocytes of different organisms [7,8,9,10,11,12,13,14].

Triglycerides are synthesized in the hepatic region from FFA and glycerol molecules. The process of triglyceride formation is regulated by several transcriptional factors such as sterol regulatory element-binding protein 1 (SREBP1c), and CCAAT/enhancer-binding protein-α (C/EBPα), which are expressed in the hepatocytes and modulate the expression of downstream genes like fatty acid synthase (FAS), responsible for fatty acid synthesis [15]. Effective diversion of the intracellular FFA towards the mitochondrial beta-oxidation process is the task of carnitine palmitoyltransferase 1A (CPT1A) [16]. For retaining normoglycemic conditions, the liver tends to metabolize excess glucose to FFA through the process of de novo lipogenesis involving activation of transcriptional factors and lipogenic genes [17]. This FFA accounts for 26% of total stored triglycerides in hepatocytes [17]. Herbal formulations have been observed to inhibit triglyceride formation through SREBP1c and C/EBPα pathways inducing lipolysis and subsequent glycerol release [15,18].

Tri-herbal decoction of Divya Sarva-Kalp-Kwath (SKK) is being used in Ayurveda for treating a plethora of liver-related ailments and is being marketed under the brand name ‘Livogrit’. SKK is prepared by combining the aqueous extracts obtained from plants identified by Council of Scientific and Industrial Research—National Institute of Science Communication and Information Resources (CSIR—NISCAIR), Delhi, India as *Boerhavia diffusa* L. (Nyctaginaceae) (NISCAIR/RHMD/Consult/2019/3453-54-149), *P. niruri* sensu Hook. f. (Euphorbiaceae) (NISCAIR/RHMD/Consult/2019/3453-54-30), and *Solanum nigrum* L. (Solanaceae) (NISCAIR/RHMD/Consult/2019/3453-54-119) plants in the ratio of 2:1:1. *B. diffusa* L. plant also known as “Punarnava” possesses a variety of isoflavonoids such as rotenoids, flavonoids, flavonoid glycosides, xanthones, lignans, ecdysteroids, and steroids [19]. These plant metabolites have been found to have hepatoprotective effects. For example, *B. diffusa* L. plant extract can modulate liver injury in animals through a reduction of cytochrome enzyme activities [20]. *P. niruri* sensu Hook. f. also known as “Bhumi amalaki” contains metabolites such as, alkaloids, anthocyanins, chlorogenic acids, flavonoids, lignans, phenolic acids, tannins, terpenoids, and saponins that attribute to its bioactivity [21,22,23,24,25,26]. *S. nigrum* L. also known as “Makoy” contains several steroidal glycosides, steroidal alkaloids and steroidal oligoglycosides that also act as antioxidants reducing hepatic injuries through amelioration of oxidative stress [27,28]. In our previous study, we performed a detailed chemical analysis of SKK by High Performance Liquid Chromatography and Liquid Chromatography based–Mass Spectroscopy. We have reported the presence of plant metabolites: gallic acid, caffeic acid, rutin, quercetin, catechin, and corilagin as marker components originating from its herbal components [19]. We have shown that SKK exhibited hepatoprotective effect in Wistar rats and HepG2 cells against Carbon Tetrachloride (CCl_4_) induced toxicity and inflammation [19].

In this study, we examined the efficacy of SKK in modulating FFA-induced cell-death and steatosis in human HepG2 cells and rat primary hepatocytes. Validation for the presence of plant metabolic components in SKK was performed using the High Performance Thin Layer Chromatography (HPTLC) method through fingerprinting of plant metabolites. Steatosis was induced in HepG2 cells and rat primary hepatocytes grown in high glucose content cell-culture media using a combination of oleic acid and palmitic acid (FFA). The onset of cell-death and steatosis by FFA in the HepG2 cells and rat primary hepatocytes and its modulation by SKK were studied through parameters such as cell viability, intracellular lipids, triglyceride accumulation, extracellular free glycerol presence, and release of liver injury biomarker aspartate aminotransferase (AST) levels. At the genetic level, C/EBPα, SREBP1c, FAS, and CPT1A mRNA expression changes induced by FFA and the ability of SKK in ameliorating these were studied in FFA induced and SKK-treated HepG2 cells.

## 2. Results

### 2.1. HPTLC Fingerprinting of Divya Sarva-Kalp-Kwath (SKK)

Preliminary screening of SKK for plant metabolites was performed using the HPTLC method (Figure 1). Results showed the presence of polyphenols namely gallic acid, caffeic acid, quercetin, catechin, rutin, and corilagin. These plant metabolites have also been identified using liquid chromatography-based mass spectroscopy (LC–MS) and high-performance liquid chromatography (HPLC) techniques in our previously published study [19]. Chromatogram analysis showed all plant metabolites of SKK were well separated without any tail or diffuseness (Figure 1A–D). Maximum retention factor (R_f_) values obtained for the different plant metabolite marker compounds were- gallic acid: 0.62 (Figure 1A, spectra A, C_1_, C_2_; Figure 1B, bands A, C_1,_ and C_2_), caffeic acid: 0.74 (Figure 1A, spectra B, C_1_, C_2_; Figure 1B, bands B, C_1,_ and C_2_), quercetin: 0.81 (Figure 1A, spectra D, C_1_, C_2_; Figure 1B, bands D, C_1_ and C_2_), catechin: 0.75 (Figure 1C, spectra E, G_1_, G_2_; Figure 1D, bands E, G_1,_ and G_2_), rutin: 0.24 (Figure 1C, spectra F, G_1_, G_2_; Figure 1D, bands F, G_1,_ and G_2_), and corilagin: 0.34 (Figure 1C, spectra H, G_1_, G_2_; Figure 1D, bands H, G_1_ and G_2_). No band intensity or position variations were observed in both replicates of SKK showing the robustness of the plant metabolites. The amount of total polyphenols present in SKK, obtained using the Folin–Ciocalteu method was 4.22% *w*/*w*. Based upon HPTLC fingerprinting and phenolic content analysis, we reaffirmed the chemical constituents of SKK.

### 2.2. SKK Modulates HepG2 Cell Viability

Preliminary cell viability screening of the SKK alone in HepG2 cells showed it to be safe up to the tested concentration of 100 µg/mL. The half-maximal inhibitory concentration (IC_50_) for SKK was calculated at 2801 µg/mL (Figure 2A). Exposure of HepG2 cells to 2 mM FFA (oleic acid (O): 1.32 mM and palmitic acid (P): 0.66 mM) induced a significant (*p* value < 0.05) loss of cell viability (70.3% ± 5.1%) after 48 h treatment (Figure 2B). Pretreatment of HepG2 cells with varying concentrations of SKK showed a significant (10 µg/mL: *p* value < 0.05 and 30 µg/mL: *p* value < 0.001) dose-dependent hepatoprotective effect against the induction of cell-death by FFA (Figure 2B). These preliminary results indicated a protective role for SKK against FFA-induced toxicity without inducing any other observable side effects like a decrease in cell viability or metabolic impairment.

### 2.3. SKK Inhibits Intracellular Lipid Accumulation

Treatment of HepG2 cells with varying combinations of oleic acid and palmitic acid showed significant (*p* value < 0.001) intracellular accumulation of lipids (2.70 ± 0.33 fold) at the 2 mM FFA, as compared to the untreated cells (Figure 3A). This was measured using neutral lipid-binding oil red ‘O’ (ORO) dye. Other combinations of oleic acid and palmitic acid did not induce any sizable changes in intracellular lipid accumulation in HepG2 cells (Figure 3A). Qualitative analysis of lipid accumulation in the HepG2 using ORO dye showed that cells treated with varying doses of SKK up to the highest therapeutic dose did not show any change in intracellular lipid accumulation compared to the normal control (Figure 3Bi,ii). However, cells treated with 2 mM FFA showed a significant increase in the intracellular lipid accumulation indicating the onset of steatosis (as seen in Figure 3Biii). Pretreatment of the HepG2 cells with varying concentrations of SKK (3, 10, and 30 µg/mL) considerably reduced the FFA-induced accumulation of intracellular lipids in the cells (Figure 3Biv,v,vi). Quantitative measurement of the ORO dye accumulation using a multiplate reader confirmed the visual observations (Figure 3B,C). Results showed that pretreatment of HepG2 cells with SKK (0, 3, 10, and 30 µg/mL) significantly (*p* value < 0.01) reduced the 2 mM FFA-induced lipid accumulation in the HepG2 cells (Figure 3C). Interestingly, SKK showed similar effects in reducing FFA-stimulated intracellular lipid accumulation at all the tested concentrations.

### 2.4. SKK Moderates Aspartate Aminotransferase (AST) Release, Intracellular Triglyceride, and Extracellular Glycerol Levels

Aspartate aminotransferase (AST) is present as cytosolic and mitochondrial isoenzyme in the liver. However, during the occurrence of hepatic steatosis, high concentrations of AST are released and can be used as a biomarker of liver injury [29]. In our study, a significant (*p* value < 0.05) increase in levels of AST levels was observed in HepG2 cells treated with 2 mM FFA (2.5 ± 0.2 fold) compared to untreated controls (Figure 4A). This FFA stimulated increase in AST level was significantly (*p* value < 0.05) reduced to normal levels upon treatment with 30 µg/mL of SKK (Figure 4A). The intracellular triglyceride synthesis requires three molecules of FFA and one molecule of glycerol [30]. Hence, the induction of steatosis in the hepatocytes is accompanied by intracellular increase in triglyceride levels and reduced extracellular levels of free glycerol molecules. HepG2 cells treated with 2 mM FFA showed a significant (*p* value < 0.05) increase in the intracellular storage of triglycerides levels (157.5 ± 49.9 µg/mg) compared to untreated cells (Figure 4A). Pretreatment of HepG2 cells with varying concentrations of SKK (0, 3, 10, and 30 µg/mL) significantly reduced the intracellular accumulation of triglycerides (52.85 ± 15.6 µg/mg) stimulated by FFA exposure (Figure 4B). Quantification of the free glycerol levels in the cell culture media, using the GC–MS method showed a significant reduction in the presence of glycerol levels in the FFA-treated HepG2 cell culture media, compared to media obtained from untreated cells, as per the area under the curve (AUC; Figure 4Ci,ii). Normalization of the glycerol contents based on individual sample protein content showed the presence of 53.07 ± 11.5 µg glycerol per mg protein in the cell culture media obtained from untreated cells. This glycerol level was reduced to 22.8 ± 4.4 µg per mg protein in the cell culture media obtained from 2 mM FFA-treated HepG2 (Figure 4Ci,ii). This represented a 0.47 ± 0.19-fold change in extracellular glycerol content (Figure 4D). Pretreatment of HepG2 cells with varying concentrations of SKK (0–30 µg/mL) significantly recovered the extracellular levels of free glycerol molecules compared to FFA exposed cells (Figure 4Ciii,iv,v,D). The results showed modulation of liver injury and lipolysis process by SKK in the hepatocytes stimulated with high FFA exposure.

### 2.5. Genomic Level Changes Induced by SKK in the In Vitro Steatosis Model

Transcription factors, fatty acid synthase, and fatty acid transporter genes play a major role in the process of FFA metabolism, triglyceride synthesis, and induction of steatosis. Treatment of HepG2 cells with 2 mM FFA induced the overexpression of steatosis associated C/EBPα (untreated cells (UC): 1.24 ± 0.82; 0 µg/mL: 9.91 ± 0.88), FAS (UC: 1.34 ± 0.94; 0 µg/mL: 4.1 ± 0.48), SREBP1c (UC: 0.61 ± 0.19; 0 µg/mL: 1.84 ± 0.53), and CPT1A (UC: 1.17 ± 0.64; 0 µg/mL: 5.43 ± 0.56; Figure 5A–D). Pretreatment of HepG2 cells with varying doses of SKK significantly ameliorated the FFA-stimulated expressions of C/EBPα, FAS, SREBP1c, and CPT1A (Figure 5A–D). However, SKK treatments did not exhibit a concentration-dependent effect. Heat-map analysis of the mRNA expression showed 2 mM FFA categorically stimulated an increase in mRNA expression levels of C/EBPα > CPT1A> FAS> SREPB1c (Figure 5E). All the FFA-stimulated mRNA expressions were back to normal levels in HepG2 cells following prophylactic treatment with SKK.

### 2.6. Intracellular Lipid Accumulation in the FFA Stimulated Rat Primary Hepatocytes

Validation of SKK’s efficacy in decreasing intracellular lipid accumulation was performed ex vivo on primary rat hepatocytes. Before analysis, a function evaluation of isolated primary hepatocytes was performed through albumin production assay. The freshly isolated rat primary hepatocytes were observed to produce 13.6 ± 0.17 mg/mL of albumin, following a 24 h ex vivo culture. Short-term (12 h) concurrent exposure of freshly isolated primary Wistar rat hepatocytes to 2 mM FFA showed a significant increase in the intracellular lipid accumulation (1.50 ± 0.01 fold) compared to untreated cells, measured using ORO dye (Figure 6). Concurrent treatment of the primary hepatocytes with SKK provided significant protection against FFA induced intracellular lipid accumulations leading to normalization of the intracellular lipid contents (Figure 6).

Overall results showed a protective effect of SKK in the human HepG2 cells and rat primary hepatocytes exposed to FFA. SKK treatment rescued the hepatocytes from excess FFA stimulated intracellular storage of lipid and triglycerides. SKK also modulated the mRNA expression of genes involved in triglyceride synthesis and intracellular storage. Finally, SKK treatment normalized the intracellular lipid and triglyceride levels of the hepatocyte, without changing their basal levels.

## 3. Discussion

Non-alcoholic fatty acid-induced hepatic steatosis is responsible for the obesity epidemic plaguing the modern world. The persistence of steatosis can lead to health-associated complications that can affect the individual quality of life. The present approach for treating free fatty acid (FFA) induced steatosis is limited to the application of thiazolidinediones and fibrates, in combination with lifestyle modifications [31]. Recent studies have shown that polyherbal formulations have equivalent efficacy to their allopathic counterparts in healing diseases without causing any side effects [32,33,34]. Therefore, herbal formulations such as Divya Sarva-Kalp-Kwath (SKK) marketed under its brand name ‘Livogrit’ is considered as an alternative therapeutic intervention against steatosis.

Plant metabolites play a major role in the disease modulating the efficacy of the herbal formulations. In our previous study, showing the efficacy of SKK formulation in ameliorating carbon tetrachloride-induced liver damages, we had identified the presence of marker plant metabolites gallic acid, caffeic acid, quercetin, catechin, rutin, and corilagin originating from the plant components *Boerhavia diffusa* L., *P. niruri* sensu Hook. f., and *Solanum nigrum* L. [19]. Hence, in the present study, the same plant metabolic compounds were analyzed using the HPTLC as a quality check for SKK. HPTLC findings indicated that these markers were amply present in the SKK formulation. Initial cell viability screening of the SKK showed them to be cytosafe in the human HepG2 cells with an IC_50_ value of 2801 µg/mL. SKK ameliorated the FFA-induced cell-death in the HepG2 cells. A similar protective response of SKK was earlier observed by us in HepG2 cells treated with carbon tetrachloride [19]. Plant metabolites comprising of rotenoids, catechin, gallic acid, quercetin, rutin, corilagin, and caffeic acid play a major role in the hepatoprotective role of SKK against a wide range of liver ailments [19,35,36,37,38].

High levels of fatty acids obtained from the diet or those produced after de novo lipogenesis increase the hepatic influx of fatty acids and stimulate ceramide synthesis. The de novo ceramide synthesis that occurs using serine and palmitoyl CoA in the endoplasmic reticulum is upregulated in the presence of an excess of saturated fatty acids. Thus, an increase in ceramide levels has been known to promote triglyceride synthesis, induce hepatic lipid accumulation, lipotoxicity, and provoke apoptosis [39,40]. Intracellular increase in the triglyceride and extracellular lipid levels decrease in the glycerol level are the hallmarks of steatosis ailment [41]. Studies on steatosis seldom emphasize the diet-related increase in hepatic lipids and remain largely focused on the transcriptional segment of fatty acid synthesis enzymes. In our present study, we selected and utilized 2 mM concentration of FFA for induction of steatosis, as other extracellular FFA concentrations did not induce a sizable fat accumulation in the HepG2 cells, under present experimental conditions. The observations are in line with the previous works of Lechón et al. and Ricchi et al. [11,13]. Therefore, this in vitro model might closely resemble the liver ailment conditions observed clinically in patients.

In our study, SKK reversed the FFA-induced steatosis conditions by reducing the intracellular lipid accumulation in both the human HepG2 cells and rat primary hepatocytes through pre- and concurrent treatments, respectively. SKK treatment also decreased the hepatocyte injury biomarker (AST) levels stimulated by FFA-induced liver damage. Our earlier study and Olaleye et al. have also shown that SKK or its component (*Boerhavia diffusa*) decreases AST levels under in vivo conditions [19,42]. In parallel, SKK also increased the presence of extracellular free glycerol molecules. These observations suggest that SKK modulated the process of triglyceride synthesis [43]. Polyphenol extracts obtained from *S. nigrum* and *P. niruri* has shown a protective effect against high fat diet-induced NAFLD in mice [21,22,28].

Hepatic lipid metabolism is a complex process involving molecular events involving the roles of transcription factors, lipogenic, and fatty acid transporter genes [44,45,46,47]. Their modulation can lead to the development of steatosis along with reduced mitochondrial energy metabolism. In our study, HepG2 cells exposed to FFA showed upregulation in the mRNA expression of SREBP1c, C/EBPα, FAS, and CPT1A genes. Prophylactic treatment of HepG2 cells with SKK downregulated these overexpressed genes. This observed downregulation of SREBP1c and C/EBPα by SKK might be directly related to the regulation of the AMPK protein phosphorylation, protecting against the progression of steatosis [43,48,49,50]. Kaempferol, a plant metabolite belonging to *B. diffusa* has been found to modulate the AMPK/mTOR pathway, an upstream regulator of SREPB-1c cleavage and activation in RIN-5F cells and murine pancreatic islets cells [51]. C/EBPα is upregulated in the liver of obese and aged mice leading to steatosis [52,53]. C/EBP-knockout in mice on a high-carbohydrate diet and obesity were found to have reduced triglyceride levels and downregulated expression of lipogenic genes [54]. Hence, in our study downregulation of the SREPB-1c and C/EBPα by SKK indicated a transcriptional level regulation of the FAA-induced steatosis.

Down-stream progression of lipogenesis is regulated by FAS, a cytosolic enzyme responsible for the synthesis of long-chained saturated fatty acids from acetyl-CoA and malonyl-CoA in the presence of NADPH [55]. Our study showed that SKK downregulated FAS overexpression in the FFA-stimulated HepG2 cells, and could be correlated to the decrease in the intracellular triglyceride storage and extracellular free glycerol level increase. Thus, SKK was able to disrupt the FAS-regulated fatty acid and triglyceride synthesis process. CPT1A is a liver isoform that catalyzes the rate-limiting steps for the transportation and conversion of fatty acids for mitochondrial beta-oxidation. Overexpression of CPT1A in FFA-treated HepG2 cells can be inferred as an adaptive response to metabolize the increased levels of available intracellular FFA. An increase in fatty acids is responsible for induction of CPT1A levels in the liver [56,57]. In the present study, CPT1A expression increased in HepG2 cells stimulated with 2 mM FFA. SKK treatment brought down this upregulated CPT1A levels to the normal level. This apparent normalization of CPT1A levels post SKK treatment points towards a chain of events leading to the attainment of homeostasis by the cells [58,59]. Prolonged enhancement of CPT1A activity can lead to increased mitochondrial energy metabolism leading to heightened reactive oxygen species generation and causing cellular damage and apoptosis [60]. Whereas, a decrease in CPT1A expression can lead to reduced β-oxidation in liver and could contribute to fatty acid accumulation and inflammation in hepatocytes [61]. Thus, our study could show that the reduced levels of intracellular lipids in both the pretreated and concurrently treated HepG2 cells and primary rat hepatocytes with SKK were through the modulation of the FFA-stimulated steatosis related transcription factors (SREBP1c and C/EBPα), and associated lipogenic gene (FAS), which in turn also normalized the fatty acid transporter (CPT1A) gene expression levels responsible for the transport of fatty acids for mitochondrial energy metabolism. Modifications in the extracellular presence of glycerol in the FFA-treated HepG2 cells pretreated with SKK further strengthen this finding. Interestingly, the regulation of intracellular lipid and triglyceride levels by SKK was close to the basal level of HepG2 cells indicating that the treatment did not impede the normal cellular metabolic activities. Quercetin is a known suppressor of SREBP1c and FAS genes involved in lipogenesis [36]. Similarly, rutin has shown therapeutic efficacy in lowering triglyceride levels and lipid droplet accumulation in preclinical models [37]. Caffeic acid is a known enhancer of lipolysis via activation of adipose triglyceride lipase activity [35]. Other flavonoids detected in SKK are known to ameliorate FFA-induced steatosis by targeting multitude of pathways involved in the development and progression of steatosis and also act additionally as antioxidant and anti-inflammatory agents [38]. We did not investigate the role of SKK alone (without FFA) in stimulating steatosis associated genes since SKK appeared benign in terms of inducing any loss of cell viability or intracellular accumulation of lipids in the HepG2 cells.

This present study model has some limitations in terms of organ-based tissue complexity and relative mode of actions. While in-vitro and ex-vivo models are good for exploring the molecular mechanisms, their results often depict signs of singularity in response. One example can be a study performed by Parra-Vargas et al. wherein they found that their test compound delphinidin used for reducing steatosis showed efficacy in vitro but not in their high-fat diet mouse model [59]. Another likely limitation is the lack of immune profiling, which is involved in the progression to fibrosis. Further, exploratory animal and human trials involving SKK will provide additional support to our observations.

In conclusion, the study outcome showed that treatment of the FFA exposed HepG2 cells and primary rat hepatocytes with tri-herbal formulation SKK (branded as Livogrit), protected against the onset and progression of hepatic lipid accumulation and steatosis. This hepatoprotective effects of SKK were facilitated through the regulation of key steatosis associated genetic elements such as transcription factors and fatty acid metabolizing/transporting genes. Furthermore, our work showed that SKK was cytosafe to hepatocytes and supported the recovery of hepatocyte viability. These results suggest that SKK may be an effective therapeutic option for treating hepatic steatosis. Further investigations would help in understanding the specific roles of plant metabolites present in SKK, in ameliorating the molecular pathology of steatosis.

## 4. Materials and Methods

### 4.1. Chemicals and Reagents

SKK was obtained from Divya Pharmacy, Haridwar, India, under its name ‘Divya Sarva-Kalp-Kwath’ (Batch no #A-SKK056). For the preparation of SKK extract (which has been branded as tablet Livogrit, by the manufacturer), we utilized the method previously described [19]. Briefly, 7.5 g of SKK powder was weighed and mixed with 400 mL of water. The mixture was boiled until a volume of 100 mL remained. The resultant decoction was dried using lyophilizer, and 853 mg of powder was obtained. Triglyceride analysis reagents were procured from Randox Laboratories Ltd., Crumlin, United Kingdom. The standards used for HPTLC were gallic acid (Sigma-Aldrich, St. Louis, MO, USA), caffeic acid (Sigma-Aldrich), quercetin (SRL, New Delhi, India), catechin (Sigma-Aldrich), rutin (Sigma-Aldrich), and corilagin (Natural remedies, Chandigarh, India). All other chemicals and reagents used for the tissue processing work were of the highest analytical grade.

### 4.2. Metabolite Analysis of SKK

Of the dried extract a 500 mg sample was dissolved in 10 mL of water followed by 20 min of sonication (normal phase, GT sonic, Guangdong, China) and centrifugation at 3000 rpm in Sorvall ST8R benchtop centrifuge (Thermo Ficher Scientific, Waltham, MA, USA) for 15 min. From the supernatant, 10 µL aliquot and 8 µL of a standard solution were loaded as 8 mm band length in a 60 F254 (10 cm × 10 cm, DA Kieselgel 60 F254, CCM Gel de silica 60 F254) plate (Merck, Kenilworth, NJ, USA) using a Hamilton syringe and ATS4 instrument (CAMAG, Muttenz, Switzerland). The samples-loaded plate was kept in TLC twin trough developing chamber (after saturation with solvent vapor) with mobile phase for chromatograph 1 (chloroform: ethyl acetate: acetone: formic acid (8: 6: 4: 2)) and chromatograph 2 (ethyl acetate: acetic acid: formic acid water (10: 1: 1: 2.3)). Development of the plate was done in the respective mobile phase up to 80 mm after which the plate was air-dried to remove the solvents. The plate was kept in a photo-documentation chamber (CAMAG visualizer) and the images were documented at UV 254 nm mode using the photo-documentation chamber. The plate was fixed and scanning was done at 280 nm by the TLC scanner. Total phenolic content of SKK was also analyzed via the Folin–Ciocalteu method using gallic acid as a standard [62].

### 4.3. In-Vitro Cell Culture

HepG2 cells were obtained from the National Centre for Cell Science, Pune, India (local repository for the American Type Cell Culture (ATCC)), proliferated within two passage numbers, and stored in liquid nitrogen at passage number 17. Cells were cultured in high glucose (4.5 gm/L) DMEM cell culture media (Gibco, Evansville, IN, USA) supplemented with 10% FBS (HiMedia, Mumbai, India) and 100 U/mL penicillin/streptomycin (Thermo Fisher Scientific, Waltham, MA, USA) and maintained at 37 °C and 5% CO_2_ [63]. Cells were passaged at 70% confluence and used for assays based on the number of cells required in each experiment. All experiments were conducted within 5 passages.

### 4.4. Cell Viability Analysis

HepG2 were plated at a density of 1 × 10^4^ cells/ well in 96 well-plate and preincubated. The cells were treated with SKK extract (0, 0.1, 0.3, 1, 3, 10, 30, 100, 300, and 1000 μg/mL) prepared in DMEM media containing 1% BSA (HiMedia, Mumbai, India) and incubated for 48 h. For the efficacy study, HepG2 cells were pretreated with SKK (0, 3, 10, and 30 μg/mL) for 24 h, followed by a combined SKK (0, 3, 10, and 30 μg/mL) and 2 mM FFA (oleic acid 1.34 mM and palmitic acid 0.66 mM) preconjugated with BSA for 48 h. Cell viability was determined using Alamar blue^®^ dye and fluorescence was measured at an excitation wavelength of 560 nm and emission wavelength 590 nm using EnVision multimode plate reader (PerkinElmer, Waltham, MA, USA).

### 4.5. Oil Red ‘O’ (ORO) Staining Based Lipid Accumulation Analysis

Intracellular lipids accumulation analysis was performed in HepG2 cells pretreated with SKK (0, 3, 10, and 30 μg/mL) for 24 h, followed by, a combined SKK (0, 3, 10, and 30 μg/mL) and 2 mM FFA (oleic acid 1.32 mM and palmitic acid 0.66 mM) pre-conjugated with BSA for 48 h. The treated cells were washed with PBS and fixed in 10% formalin for 30 min. A 0.6% (*w/v*) stock solution of ORO was prepared in 2-propanol (IPA) and filtered through a 0.22 μm membrane filter. A working solution of ORO was prepared by mixing 3 parts of the stock solution and 2 parts of ddH_2_O. The fixed HepG2 cells were stained with the ORO working solution for 40 min at room temperature. The stained HepG2 cells were washed four times with ddH2O to remove unbound stain and qualitatively analyzed using a Zeiss Primovert inverted bright-field microscope. For lipid quantification, 100% of IPA was added to ORO stained HepG2 cells and the dissolved cellular stain was measured at an absorbance wavelength of 518 nm using an EnVision multimode plate reader (PerkinElmer, Waltham, MA, USA).

### 4.6. Intracellular Triglyceride Estimation

HepG2 cells were plated at a density of 1 × 10^5^ cells/well in a 12-well plate and preincubated overnight. The cells were pretreated with a varying dose of SKK (0, 3, 10, and 30 μg/mL), followed by a combination of 2 mM FFA and SKK (0, 3, 10, and 30 μg/mL) for a period of 48 h. On the completion of the exposure period, exposed cell culture media was collected for extracellular glycerol analysis, and cells were washed thrice with PBS and treated with ice-cold (4 °C) PBS. The cells were scraped using a cell-scraper and lysed after two freeze–thaw cycles in liquid nitrogen. Triglyceride analysis in the cell lysate was done using the Randox triglyceride assay kit and Randox Monaco clinical chemistry analyzer (Randox Laboratories Ltd., Crumlin, UK). The analysis was performed following the manufacturer’s guidelines. Total protein content of the individual samples was measured using the bicinchoninic acid (BCA) colorimetric kit (Thermo Fisher Scientific, Waltham, MA, USA) and the measurement was obtained at 562 nm using EnVision plate reader (PerkinElmer, Waltham, MA, USA). Measured levels of proteins per sample were used for normalization of the triglyceride levels.

### 4.7. AST Estimation

Cells were plated at a density of 1 × 10^5^ cells/ well in a 12-well plate and incubated overnight. The cells were pretreated with a varying dose of SKK (0–30 μg/mL), followed by a combination of 2 mM FFA and SKK (0–30 μg/mL) for a period of 48 h post which the supernatant was collected and centrifuged at 100× *g* for 5 min to remove cell debris. The supernatant was then transferred to 1.5 mL vials of Biochemical analyzer XL 640 (Erba Mannheim, Czech Republic). The analysis was performed following the manufacturer’s guidelines. Total protein content of the sample was analyzed using the BCA kit (mentioned above) and data were normalized using obtained values.

### 4.8. Estimation of Extracellular Free Glycerol

The cell culture media collected from the triglyceride assay (see above) was centrifuged at 100× *g* for 5 min for removing the cell debris. The supernatant was collected and transferred to 2 mL vial for gas chromatography–mass spectroscopy (GC–MS). The analysis was performed on GC–MS (7000D GC/MS triple quad attached with the 7890B GC system (Agilent Technologies, Santa Clara, CA, USA) instrument. Data acquisition and analysis was performed using mass hunter software. For separation, the DB-624 capillary column (30 m × 0.25 mm, 0.25 µm) was utilized and helium was used as carrier gas with a flow-rate of 1.5 mL/min. The temperature of the injector was set at 280 °C and the split ratio was adjusted to 5:1. The column temperature was set at 80 °C (with 2 min hold) and programmed at 5 °C/min to 150 °C (with 2 min hold), followed by 10 °C/min to 200 °C (with 2 min hold). The temperature of the GC–MS ion source was 240 °C and its ionization potential was 70 eV. The spectra were acquired using the MS1 SIM mode with selected ion for glycerol (Quantifier-61 and Qualifier-43). A standard curve was prepared using glycerol concentrations of 10, 20, 50, 100, and 200 ppm. The area under the curve was analyzed for each spectrum of glycerol for quantification purposes. Total protein content of the sample was analyzed using BCA kit (mentioned above), which was used for the normalization of glycerol levels per sample.

### 4.9. Real-Time PCR (RT-PCR) Based Gene Expression Quantification

Total RNA was isolated from HepG2 after treatment using the RNeasy mini kit (Qiagen, Hilden, Germany) following the manufacturer’s protocol. The cDNA synthesis was done using the Verso cDNA synthesis kit (Thermo Fisher Scientific, Waltham, MA, USA). cDNA samples were mixed with the PowerUp SYBR Green Master Mix (Applied Biosystems, Foster city, CA, USA) and RT-PCR was performed using qTOWER^3^ G (Analytik-Jena, Jena, Germany). The RT-PCR cycling parameters included an initial denaturation of 95 °C for 10 min and a primer extension at 95 °C for 15 s and 60 °C for one minute with 40 cycles. Ct values were obtained, and relative expression 2^(−ΔΔCt)^ was calculated and analyzed for changes in mRNA expression. Primers used for the study are mentioned in Table 1. Peptidylprolyl Isomerase A (PPIA) gene was used as a housekeeping gene.

### 4.10. Primary Rat Hepatocyte Isolation and Treatment

Male Wistar rats (250–300 gm) were procured from Hylasco Bio-Technology Pvt. Ltd. (Hyderabad, India). All the animals were placed under a controlled environment with a relative humidity of 60–70% and 12:12 h light and dark cycle in a registered animal house (1964/PO/RC/S/17/CPCSEA) of Patanjali Research Institute, Haridwar, India. The animals were fed a standard pellet diet (Purina Lab Diet, St. Louis, MO, USA) and sterile filtered water ad libitum. The animals used for this study were part of protocol approved by the Institutional Animal Ethical Committee (IAEC) of Patanjali Research Institute vide IAEC approval number: LAF/PCY/2020-39.

Isolation of primary hepatocytes was isolated from the liver tissues of three Wistar rats separately following the protocol mentioned by Shen et al., 2012 with slight modification [64]. Briefly, liver was cut into 1 mm thick pieces and perfused with hepatocyte wash medium (Gibco, Evansville, IN, USA). These liver pieces were placed in 0.25% Trypsin + EDTA solution (Invitrogen, Carlsbad CA, USA) for 30 min till transformation into soft mushy texture. The whole tissue was then forced to pass through 100–40-micron sieves, serially. The filtrate was added to 30 mL of complete DMEM medium and centrifuged at 50× *g*. The pellet was collected and cell viability of the isolated hepatocytes was measured up to 65% using trypan blue-based assay. Cells were plated in 96 well plates at a cell density of 2 × 10^4^ cells/well and maintained in normal Williams’ media E (Gibco, Evansville, IN, USA) and Opti MEM (Gibco, Evansville, IN, USA) mixed in 1:1 ratio supplemented with 5% FBS and 2% antibiotic at 37 °C and 5% CO_2_. For functional assay, cells were plated in a 12-well plate at a density of 3 × 10^5^ cells/well. Albumin was analyzed post 24 h of culture from the supernatant of plated cells using commercially available kits for Albumin (Randox Laboratories Ltd., Crumlin, UK) and measured using Randox Monaco clinical chemistry analyzer (Randox Laboratories Ltd., Crumlin, UK), following the manufacturer’s instructions. The primary hepatocytes were treated with a combination of 2 mM FFA along with varying concentrations of SKK (0, 3, 10, and 30 µg/mL) and incubated for 12 h, following which the intracellular lipid accumulation was evaluated using the ORO assay.

### 4.11. Statistical Analysis

All the experiments were repeated thrice in triplicate. Data represents mean ± standard error of mean (SEM) over 95% confidence intervals. Statistical analysis was performed using one-way ANOVA with Dunnett’s multiple comparisons post-hoc test. Statistical analysis was performed using GraphPad Prism 7 (GraphPad Software, Inc., San Diego, CA, USA). The results were considered to be statistically significant at a probability level of *p* value < 0.05.

## Figures and Tables

**Figure 1 molecules-25-04849-f001:**
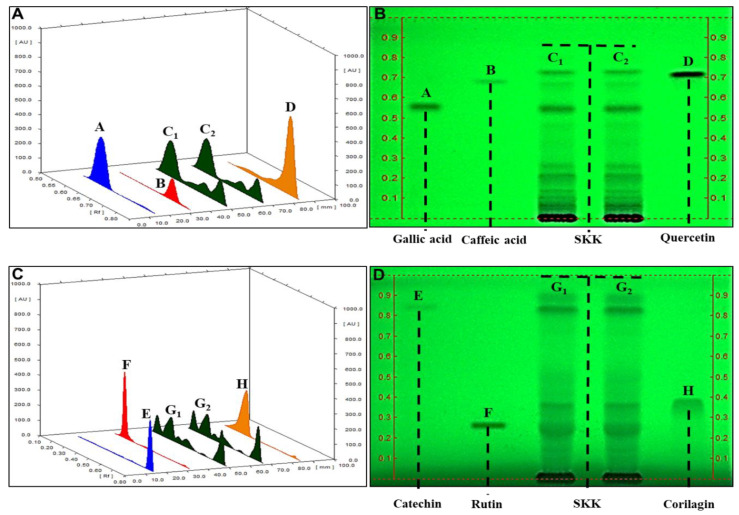
Chromatograms of Sarva-Kalp-Kwath (SKK) extract in the High Performance Thin Layer Chromatography (HPTLC) analysis. Chemical constituent analysis of SKK using HPTLC showed (**A**) comparative fingerprint analysis at 280 nm of (C_1_, C_2_) SKK with (A) gallic acid, (B) caffeic acid, and (D) quercetin markers. (**B**) Corresponding comparative HPTLC image at 254 nm of (C_1_, C_2_) SKK with (A) gallic acid, (B) caffeic acid, and (D) quercetin markers. (**C**) Comparative fingerprint at 280 nm of (G_1_, G_2_) SKK with (E) catechin, (F) rutin, and (H) corilagin markers. (**D**) Corresponding comparative HPTLC image at 254 nm of (G_1_, G_2_) SKK with (E) catechin, (F) rutin, and (H) corilagin markers. Results are displayed as 3D-overlaid spectra of SKK and marker compounds and their respective HPTLC chromatograms.

**Figure 2 molecules-25-04849-f002:**
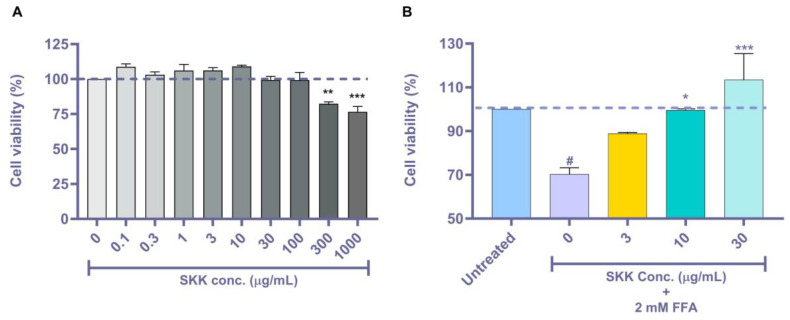
Effect of SKK on HepG2 cell viability. (**A**) Cytosafety profiling of HepG2 cells post-SKK (0–1000 μg/mL) treatment. (**B**) Cell viability analysis of free fatty acid (FFA)-treated HepG2 cells post-SKK (0, 3, 10, and 30 μg/mL) treatment. All the experiment was performed thrice in triplicate and the results are displayed as mean ± SEM. For statistical analysis, one-way ANOVA with Dunnett’s multiple comparisons was done, wherein A) SKK doses vs. 0 μg/mL ** *p* value < 0.01 and *** *p* value < 0.001 and in (**B**) untreated control vs. 0 μg/mL= # *p* value < 0.05, and SKK test concentrations vs. 0 μg/mL= * *p* value < 0.05 and *** *p* value < 0.001.

**Figure 3 molecules-25-04849-f003:**
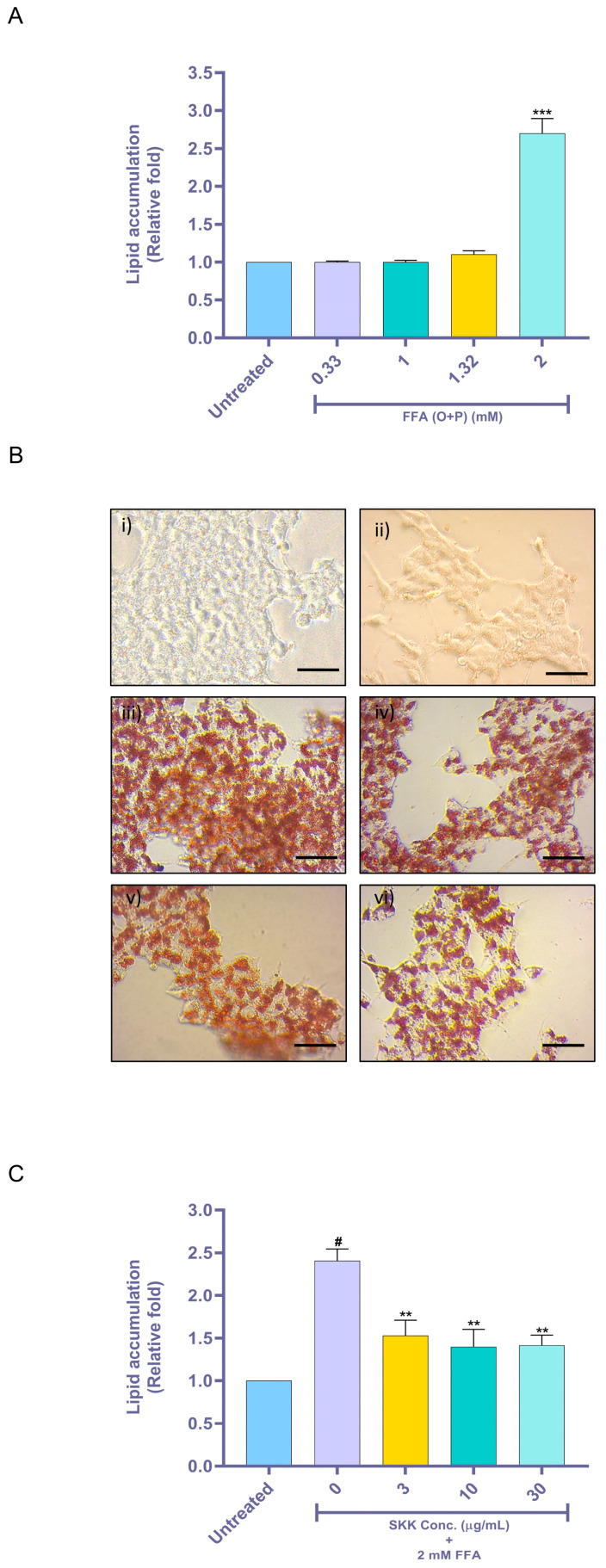
Intracellular lipid accumulation and its amelioration by SKK. (**A**) Intercellular lipid accumulation was evaluated in HepG2 cells following treatment with varying combinations of oleic acid (O) and palmitic acid (P). (**B**) Visual analysis for oil red ‘O” dye accumulation associated with neutral lipids was performed in the HepG2 cells following pretreatment with SKK (0, 3, 10, and 30 μg/mL) for 24 h, and thereafter a combined treatment of SKK (0, 3, 10, and 30 μg/mL), and 2 mM FFA for 48 h. The scale bar in the images represents 20 µm. (**C**) ORO staining results showed that SKK pretreatment inhibited the intracellular lipid accumulation stimulated by FFA exposure in the HepG2 cells. All the experiments were performed thrice in triplicate and the results are displayed as mean ± SEM. For statistical analysis, one-way ANOVA with Dunnett’s multiple comparisons was done, where untreated vs. FFA= *** *p* value < 0.001, untreated control vs. 0 μg/mL= # *p* value < 0.001, and SKK test concentrations vs. 0 μg/mL= ** *p* value < 0.01.

**Figure 4 molecules-25-04849-f004:**
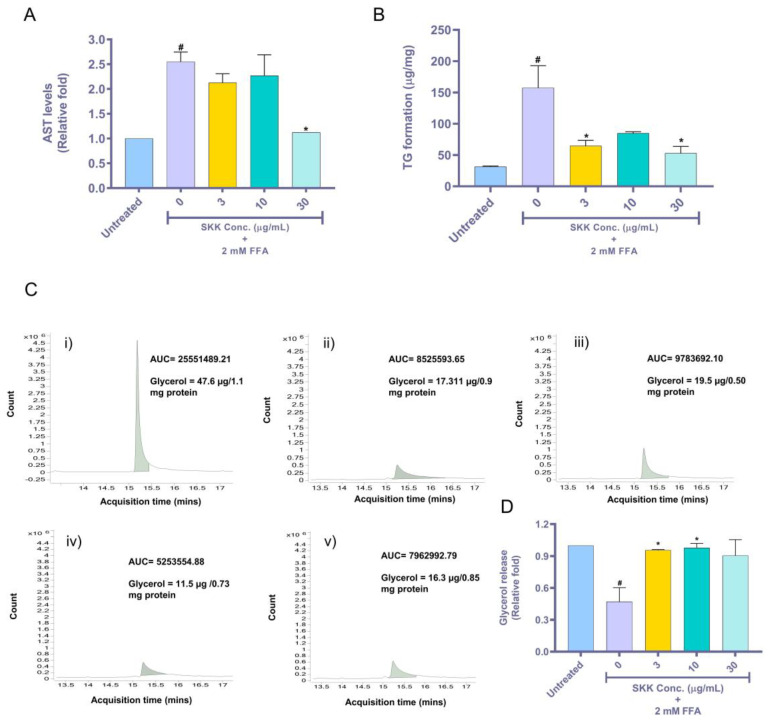
Quantification of intracellular triglyceride and extracellular glycerol release. (**A**) AST levels and (**B**) triglyceride accumulation, in HepG2 following treatment with SKK (0, 3, 10, and 30 μg/mL). (**C**) The area under the curve (AUC) measurements obtained from the GC–MS analysis of the cell culture media for the presence of free glycerol. Spectra represents the presence of glycerol in cell culture media obtained from Untreated cells (**i**), 2 mM FFA-stimulated HepG2 cells (**ii**), HepG2 cells pretreated with 3 µg/mL SKK (**iii**), 10 µg/mL SKK (**iv**), and 30 µg/mL SKK (**v**) followed by 2 mM FFA and SKK (respected concentration) for 48 h. The representative spectra show AUC representing the presence of free glycerol molecules in the cell culture media. (**D**) Relative fold changes measured in the quantity of free glycerol levels present in the cell culture media obtained from untreated, FFA-treated, and SKK pretreated and FFA and SKK cotreated cells, normalized with individual protein levels in each sample. All the experiments were performed thrice in triplicate, and samples were pooled into 3 biological replicate groups for analysis purposes. Results are displayed as mean ± SEM. For statistical analysis, one-way ANOVA with Dunnett’s multiple comparisons was performed, where untreated control vs. 0 μg/mL= # *p* value < 0.001 and SKK test concentrations vs. 0 μg/mL= * *p* value < 0.05.

**Figure 5 molecules-25-04849-f005:**
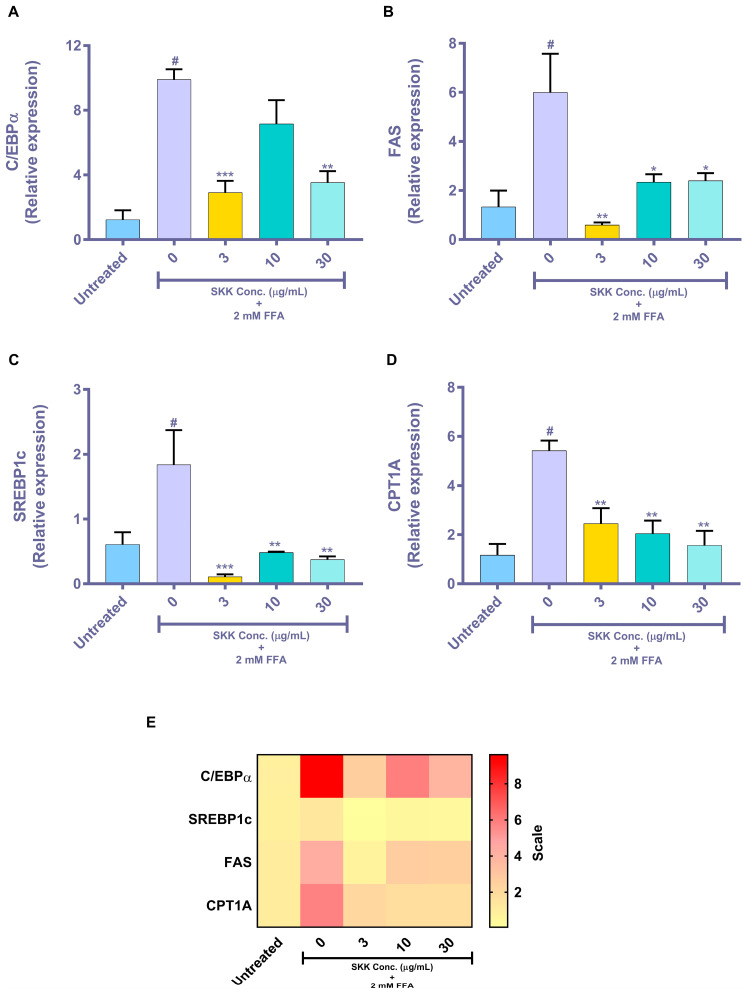
Gene expression analysis in HepG2 cells treated with free fatty acids and SKK. FFA-stimulated overexpression of steatosis associated genes: (**A**) C/EBPα, (**B**) FAS, (**C**) SREBP1c, and (**D**) CPT1A were observed in HepG2 cells. Pretreatment with SKK ameliorated the overexpression of genes induced by FFA treatment. (**E)** Heat-map presents a visual perspective of the up- and down-regulated genes in the 2 mM FFA and SKK treated HepG2 cells. Results are displayed as mean ± SEM. All the experiments were performed thrice in duplicate. For statistical analysis, one-way ANOVA with Dunnett’s multiple comparisons was performed, where untreated control vs. 0 μg/mL= # *p* value < 0.01 and SKK test concentrations vs. 0 μg/mL= * *p* value < 0.05, ** *p* value < 0.01 and *** *p* value < 0.001.

**Figure 6 molecules-25-04849-f006:**
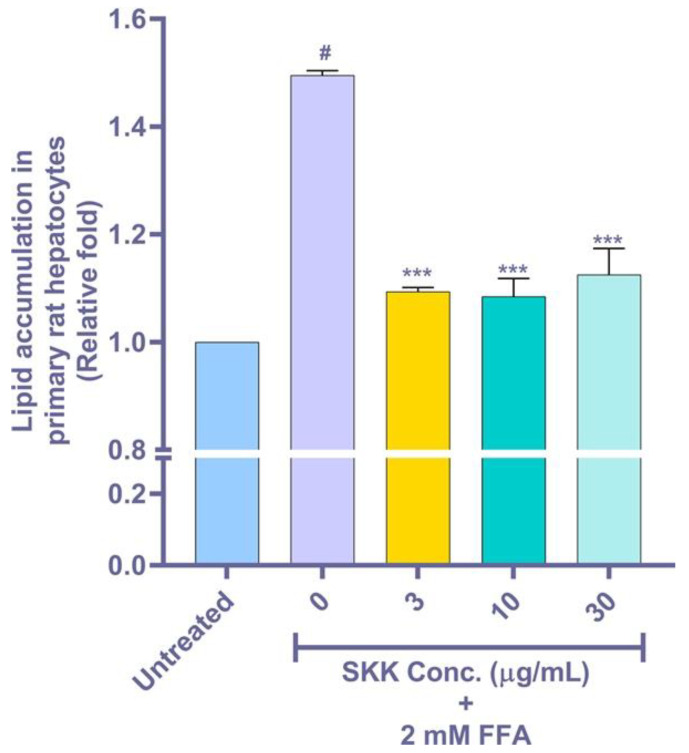
Lipid accumulation and its amelioration by SKK in primary rat hepatocytes. Steatosis induction was evaluated using isolated rat hepatocytes after induction with 2 mM FFA mixture and treatment with SKK (0, 3, 10 and 30 µg/mL) for 12 h. Results are displayed as mean ± SEM. The experiment was performed thrice in duplicate. For statistical analysis, one-way ANOVA with Dunnett’s multiple comparisons was performed, where untreated control vs. 0 μg/mL= # *p* value < 0.001 and SKK test concentrations vs. 0 μg/mL= *** *p* value < 0.001.

**Table 1 molecules-25-04849-t001:** Primers used for the analysis of gene expression.

Gene Name	Primer Sequence
C/EBPα	F: 5′-TGGACAAGAACAGCAACGAGTA-3′ R: 5′-ATTGTCACTGGTCAGCTCCAG-3′
SREBP1c	F: 5′-GCGCCTTGACAGGTGAAGTC-3′ R: 5′-GCCAGGGAAGTCACTGTCTTG-3′
FAS	F: 5′-CCCCTGATGAAGAAGGATCA-3′ R: 5′-ACTCCACAGGTGGGAACAAG-3′
CPT1A	F: 5′-CCTCCGTAGCTGACTCGGTA-3′ R: 5′-GGAGTGACCGTGAACTGAAA-3′
PPIA	F: 5′-CCCACCGTGTTCTTCGACATT-3′ R: 5′-GGACCCGTATGCTTTAGGATGA-3′

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
