# Peer review of "Tri-Herbal Medicine Divya Sarva-Kalp-Kwath (Livogrit) Regulates Fatty Acid-Induced Steatosis in Human HepG2 Cells through Inhibition of Intracellular Triglycerides and Extracellular Glycerol Levels"

_molecules, 2020, doi:10.3390/molecules25204849_

Round 1
Reviewer 1 Report
The manuscript entitled “Tri-herbal decoction Divya Sarva-Kalp-Kwath regulates fatty acid-induced steatosis in human liver (HepG2) cells through inhibition of intracellular triglycerides and extracellular glycerol levels” aims to explore the in vitro effect of the tri-herbal decoction Divya Sarva-Kalp-Kwath (SKK) on HepG2 cells and rat primary hepatocytes also examining the possible molecular mechanism involved. Overall, the manuscript is clearly written, and experiments were well conducted. However, there are several concerns that the authors should address in a point-by-point fashion before the article can be accepted for publication.
Material and Methods
- Please provide more details regarding HepG2 cell in vitro cultures (i.e., number of previous passages, percentage of confluence for in vitro experiments, etc).
- As the authors describe, the culture medium used for in vitro experiments is a high glucose culture medium. High glucose is well known as a hepatic steatosis inductor both in vivo and vitro. So, the possible synergistic effects of high glucose and free fatty acid (FFA) exposure on HepG2 cells are uncertain in this work and should be discussed.
- It is hard to believe you were able to isolate rat primary hepatocytes without using collagenase. Did you confirm that isolated cells were indeed hepatocytes (i.e., by performing western blot for albumin)? What was the cell viability of these cells after being isolated? Why these cells were incubated for 12 h and HepG2 cells for 24 h?
- Is n=3 enough to perform solid statistical analysis? Please provide the rationale behind using n=3 in most of the experiments.
- Considering n=3, how was data distribution? Was a normal distribution? Did you perform one-way ANOVA one or two-tailed?
Results
- It is quite impressive the significant increase in cell viability in HepG2 cells treated with 30 ug/ml SKK. Considering that alamarBlue Assay was not only designed to measure cell viability but also cell proliferation, it is feasible to suppose that 30 ug/ml SKK might promote hepatocyte proliferation, at least on in vitro culture. I think the authors should discuss this potential side-effect of SKK (it is important when you discuss about using SKK to treat patients with liver steatosis that are at higher risk of developing cirrhosis and hepatocellular carcinoma).
- Why the authors present triglyceride content as relative fold and not in ug per mg protein?
- How do you explain that 10 ug/ml SKK does not change C/EBP expression but 3 and 30 ug/ml SKK are able to decrease it?
- C/EBP, FAS, and SREBP1c are involved in lipogenesis. In contrast, CPT1A is involved in lipolysis. I still cannot understand why CPT1A shows the same behavior than C/EBP, FAS, and SREBP1c in response to SKK, even in the presence of 2 mM FFA.
- A major limitation of this study is that C/EBP, FAS, SREBP1c, and CPT1A were only measured by qPCR and not western blot. In my opinion, it is extremely important to determine protein levels of the aforementioned molecules with the aim of fully elucidating their role in preventing lipid accumulation in response to SKK.
Discussion
- Discussion limits to present results again without comparing the main results with current literature.
- The most interesting section of the discussion (lines 279-303) is highly speculative due to the authors provide information regarding the activity of C/EBP, FAS, SREBP1c, and CPT1A without clarifying they only measured mRNA expression.
- Please discuss your own findings in light of contradictory studies that show no real benefits of herbal formulations in steatosis.
- Please include the possible effect of SKK on hepatocyte proliferation as a potential side-effect.
- Include limitations of the study.
Author Response
Reviewer 1
Query 1: Please provide more details regarding HepG2 cell in vitro cultures (i.e. number of previous passages, percentage of confluence for in vitro experiments, etc).
Response: We thank the reviewer for the query. Based on the suggestion, we have now added additional information regarding the passage number, confluency, and cell culture maintenance.
Page 13, Line 396: Material and Methods
HepG2 cells were obtained from the National Centre for Cell Science, India (local repository for the American Type Cell Culture (ATCC)), proliferated within two passage numbers, and stored in liquid nitrogen at passage number 17. Cells were cultured in high glucose (4.5 gm/L) DMEM cell culture media (Gibco, USA) supplemented with 10% FBS (HiMedia, India) and 100 U/ml penicillin/streptomycin (Thermo Fisher Scientific, Waltham, MA, USA) and maintained at 37°C and 5% CO2 [49]. Cells were passaged at 70% confluence and used for assays based on the number of cells required in each experiment. All experiments were conducted within 5 passages.
Reference.
Fan, H.; Chen, Y.Y.; Bei, W.J.; Wang, L.Y.; Chen, B.T.; Guo, J. In Vitro Screening for Antihepatic Steatosis Active Components within Coptidis Rhizoma Alkaloids Extract Using Liver Cell Extraction with HPLC Analysis and a Free Fatty Acid-Induced Hepatic Steatosis HepG2 Cell Assay. Evid Based Complement Alternat Med 2013, 2013, 459390, doi:10.1155/2013/459390.
Query 2: As the authors describe, the culture medium used for in vitro experiments is a high glucose culture medium. High glucose is well known as a hepatic steatosis inductor both in vivo and in vitro. So, the possible synergistic effects of high glucose and free fatty acid (FFA) exposure on HepG2 cells are uncertain in this work and should be discussed.
Response: We thank the reviewer for the keen observation. The media composition was selected based on the previous work by Fan et al., 2013. It is well known that both high carbohydrate and lipid diet results in hepatic steatosis. In this context, we have added in the manuscript-
Page 2, Line 61: Introduction
For retaining normoglycemic conditions, the liver tends to metabolize excess glucose to FFA through the process of de novo lipogenesis involving activation of transcriptional factors and lipogenic genes [8]. This FFA accounts for 26% of total stored triglycerides in hepatocytes [8].
Reference.
Fan, H.; Chen, Y.Y.; Bei, W.J.; Wang, L.Y.; Chen, B.T.; Guo, J. In Vitro Screening for Antihepatic Steatosis Active Components within Coptidis Rhizoma Alkaloids Extract Using Liver Cell Extraction with HPLC Analysis and a Free Fatty Acid-Induced Hepatic Steatosis HepG2 Cell Assay. Evid Based Complement Alternat Med 2013, 2013, 459390, doi:10.1155/2013/459390.
Donnelly, K.L.; Smith, C.I.; Schwarzenberg, S.J.; Jessurun, J.; Boldt, M.D.; Parks, E.J. Sources of fatty acids stored in liver and secreted via lipoproteins in patients with nonalcoholic fatty liver disease. J Clin Invest 2005, 115, 1343-1351, doi:10.1172/JCI23621.
Query 3: It is hard to believe you were able to isolate rat primary hepatocytes without using collagenase. Did you confirm that isolated cells were indeed hepatocytes (i.e., by performing western blot for albumin)? What was the cell viability of these cells after being isolated? Why these cells were incubated for 12 h and HepG2 cells for 24 h?
Response: We thank the reviewer for the query and agree that it is indeed important to identify the primary rat hepatocytes we have isolated.
- In this context, firstly we did not use collagenase for the isolation of primary hepatocytes from freshly isolated rat liver. We applied the isolation protocol described by Shen et al., 2012. In the manuscript, we have now updated the isolation and cell culture conditions as-
Page 16, Line 476: Material and Methods
Male Wistar rats (250-300 gm) were procured from Hylasco Bio-Technology Pvt. Ltd (Hyderabad, India). All the animals were placed under a controlled environment with a relative humidity of 60–70% and 12:12 hr light and dark cycle in a registered animal house (1964/PO/RC/S/17/CPCSEA) of Patanjali Research Institute, India. The animals were fed a standard pellet diet (Purina Lab Diet, St. Louis, MO, USA) and sterile filtered water ad libitum. The animals used for this study were part of protocol approved by the Institutional Animal Ethical Committee (IAEC) of Patanjali Research Institute vide IAEC approval number: LAF/PCY/2020-39.
Isolation of primary hepatocytes was isolated from the liver tissues of three Wistar rats separately following the protocol mentioned by Shen et al., 2012 with slight modifications [50]. Briefly, liver was cut into 1 mm thick pieces and perfused with hepatocyte wash medium (Gibco, Thermo Scientific). These liver pieces were placed in 0.25 % Trypsin + EDTA solution (Invitrogen) for 30 min. as till transformation into soft mushy texture. The whole tissue was then forced to pass through 100 to 40-micron sieves, serially. The filtrate was added to 30 ml of complete DMEM medium and centrifuged at 50×g. The pellet was collected and the cell viability of the isolated hepatocytes was measured upto 65% using trypan blue based assay. Cells were plated in 96 well plates at a cell density of 2 × 104 cells/well and maintained in normal Williams’ media E (Gibco, India) and Opti MEM (Gibco, India) mixed in 1:1 ratio supplemented with 5% FBS and 2% antibiotic at . 37°C and 5% CO2.
Reference.
Shen, L.; Hillebrand, A.; Wang, D.Q.; Liu, M. Isolation and primary culture of rat hepatic cells. J Vis Exp 2012, 10.3791/3917, doi:10.3791/3917.
- For functional confirmation of the hepatocytes, we did perform an albumin assay. This has now been included in the manuscript as-
Page 16, Line 492: Material and Methods
For functional assay, cells were plated in a 12-well plate at a density of 3 × 105 cells/well . Albumin was analyzed post 24 hr of culture from the supernatant of plated cells using commercially available kits for Albumin (AB8301, Randox, UK) and measured using Randox Monaco clinical chemistry analyzer (Randox, UK), following manufacturer’s instructions.
Page 8, Line 222: Results
Before analysis, a function evaluation of isolated primary hepatocytes was performed through albumin production assay. The freshly isolated rat primary hepatocytes were observed to produce 13.6±0.17 mg/ml of albumin, following a 24 hrs ex vivo culture.
- Compared to the HepG2 cells, the primary rat hepatocytes showed significant intracellular lipid droplet accumulation within 6 hrs following incubation with FFA (Figures 1 and 2). Therefore, 12 hr incubation was selected as the optimum time for studying the efficacy of SKK in ameliorating FFA induced steatosis.
Figure 1: Normal control untreated rat primary hepatocytes showing absence of intracellular lipid droplets; Figure 2: Accumulation of lipid droplets in rat primary hepatocytes following treatment with FFA.
Query 4: Is n=3 enough to perform solid statistical analysis? Please provide the rationale behind using n=3 in most of the experiments.
Response: We thank the reviewer for pointing us the mistake. We have now replaced ‘n=3’ with the following statement, ‘All the experiments were performed are thrice in triplicates’ (3 biological and 3 technical) throughout the manuscript.
Query 5: Considering n=3, how was data distribution? Was a normal distribution? Did you perform one-way ANOVA one or two-tailed?
Response. As mentioned in response to the query above, all the experiments were performed thrice in triplicate (now corrected in manuscript) consisting of 3 technical replicates and 3 biological replicates. Individual sets of experiments were completed within 5 passages of the cells. The inter- and intra-experimental data distribution was within 10% of the Coefficient of Variation. For the study, we had performed one-way ANOVA with Dunnett’s posthoc modification.
Query 6: Results. It is quite impressive the significant increase in cell viability in HepG2 cells treated with 30 ug/ml SKK. Considering that Alamar Blue Assay was not only designed to measure cell viability but also cell proliferation, it is feasible to suppose that 30 ug/ml SKK might promote hepatocyte proliferation, at least on in vitro culture. I think the authors should discuss this potential side-effect of SKK (it is important when you discuss about using SKK to treat patients with liver steatosis that are at higher risk of developing cirrhosis and hepatocellular carcinoma).
Response. We again thank the reviewer for the keen observation. However, we would like to point out that the SKK treatment of HepG2 cells with SKK alone did not induce any proliferative activity (see Figure 2A in manuscript). The IC50 value for SKK was observed to be 2801 µg/ml. HepG2 cells treated with the 2:1 combination of Oleic acid (O) and Palmitic acid (P), and 30 µg/ml of SKK showed an increase of 113.4 ± 12.07% in cell viability activity, which is well within the acceptable range (see Figure 2B). In addition, we have earlier repoted that SKK formulation did not induce any toxicity up to 1000 mg/kg/day in 28 day repeated dose sub-acute Tox study, in wistar rats (Balkrishna et al., 2020).
Reference.
- Balkrishna et al., “Polyherbal Medicine Divya Sarva-Kalp-Kwath Ameliorates Persistent Carbon Tetrachloride Induced Biochemical and Pathological Liver Impairments in Wistar Rats and in HepG2 Cells,” Front. Pharmacol., vol. 11, Mar. 2020, doi: 10.3389/fphar.2020.00288.
Query 7: Why the authors present triglyceride content as relative fold and not in ug per mg protein?
Response: We thank the reviewer for the excellent suggestion. We have now converted the triglyceride levels based on cellular protein content into µg/mg of protein (Figure 4B).
Query 8: How do you explain that 10 ug/ml SKK does not change C/EBP expression but 3 and 30 ug/ml SKK are able to decrease it?
Response: Thank you for pointing out the dose variation in the C/EBP results. We agree that the 10 µg/mL treatment of SKK did show a significant down-regulation of the FFA stimulated C/EBP activity as compared to the other two doses. This was repeated in all the biological and technical replicates. While the response was lower than the control, it is difficult to explain at this point. We may speculate that the middle dosage might have a mild antagonistic effect thereby stimulating C/EBP response in the presence of the FFA. We will explore this possibility in our future research work.
Query 9: C/EBP, FAS, and SREBP1c are involved in lipogenesis. In contrast, CPT1A is involved in lipolysis. I still cannot understand why CPT1A shows the same behavior as C/EBP, FAS, and SREBP1c in response to SKK, even in the presence of 2 mM FFA.
Response. Thank you for this idea. However, our study results showed that SKK was capable of modulating CPT1A expression back to normal levels, even in the presence of high fat and carbohydrate contents. It was infact not down-regulated. This is indeed a good indication as SKK did not reduce FFA metabolism below normal levels, that may lead to cellular energy starvation. Furthermore, it is well known that prolonged enhancement of CPT1A activity can lead to increased mitochondrial energy metabolism leading to heightened reactive oxygen species generation causing cellular damages and apoptosis (Brown et al., 2018).
Reference
Brown ZJ, Fu Q, Ma C, Kruhlak M, Zhang H, Luo J, Heinrich B, Yu SJ, Zhang Q, Wilson A, Shi ZD, Swenson R, Greten TF. Carnitine palmitoyltransferase gene upregulation by linoleic acid induces CD4+ T cell apoptosis promoting HCC development. Cell Death Dis. 2018 May 23;9(6):620. doi: 10.1038/s41419-018-0687-6. PMID: 29795111; PMCID: PMC5966464.
Query 10: A major limitation of this study is that C/EBP, FAS, SREBP1c, and CPT1A were only measured by qPCR and not western blot. In my opinion, it is extremely important to determine protein levels of the aforementioned molecules with the aim of fully elucidating their role in preventing lipid accumulation in response to SKK.
Response. This is indeed an excellent remark. However, the expression levels of our gene of interest were correlated to their respective western blots in a similar study done by Park et al., 2019 (as shown in the figure below) wherein the protein level of lipogenic factors, SREBP-1c, C/EBPα, FAS were correlating with their respective expression profiles. So, we sought not to replicate these findings.
Reference.
Park, M.; Yoo, J.H.; Lee, Y.S.; Lee, H.J. Lonicera caerulea Extract Attenuates Non-Alcoholic Fatty Liver Disease in Free Fatty Acid-Induced HepG2 Hepatocytes and in High Fat Diet-Fed Mice. Nutrients 2019, 11, doi:10.3390/nu11030494.
Query 11: Discussion
- Discussion limits to present results again without comparing the main results with current literature.
- The most interesting section of the discussion (lines 279-303) is highly speculative due to the authors provide information regarding the activity of C/EBP, FAS, SREBP1c, and CPT1A without clarifying they only measured mRNA expression.
- Please discuss your own findings in light of contradictory studies that show no real benefits of herbal formulations in steatosis.
- Please include the possible effect of SKK on hepatocyte proliferation as a potential side-effect. Include limitations of the study.
Response:
- We appreciate the suggestions of the reviewer that more preference should be given to the correlation of our study findings with existing literature and we have complied with the suggestion in the discussion section.
- We specifically have mentioned in the discussion that mRNA expression was used for stating the inference of our observations.
- Findings contradictory to our study showing disparity in the effect of herbal formulations in steatosis has been added in discussion section.
Page 11, Line 354: Discussion
One example can be a study performed by Parra-Vargas et al wherein they found that their test compound delphinidin used for reducing steatosis showed efficacy in vitro but not in their high high-fat diet mouse model [46].
Reference.
Parra-Vargas, M.; Sandoval-Rodriguez, A.; Rodriguez-Echevarria, R.; Dominguez-Rosales, J.A.; Santos-Garcia, A.; Armendariz-Borunda, J. Delphinidin Ameliorates Hepatic Triglyceride Accumulation in Human HepG2 Cells, but Not in Diet-Induced Obese Mice. Nutrients 2018, 10, doi:10.3390/nu10081060.
- The limitations of the study have been added to the manuscript.
Page 11, Line 352: Discussion
The present study model has some limitations in terms of organ-based tissue complexity and relative mode of actions. While in vitro and ex vivo models are good for exploring the molecular mechanisms, their results often depict signs of singularity in response. One example can be a study performed by Parra-Vargas et al wherein they found that their test compound delphinidin used for reducing steatosis showed efficacy in vitro but not in their high high-fat diet mouse model [46]. Another likely limitation is the lack of immune profiling which is involved in progression to fibrosis. Further, exploratory animal and human trials involving SKK will provide additional support to our observations.
Reference.
Parra-Vargas, M.; Sandoval-Rodriguez, A.; Rodriguez-Echevarria, R.; Dominguez-Rosales, J.A.; Santos-Garcia, A.; Armendariz-Borunda, J. Delphinidin Ameliorates Hepatic Triglyceride Accumulation in Human HepG2 Cells, but Not in Diet-Induced Obese Mice. Nutrients 2018, 10, doi:10.3390/nu10081060.

Reviewer 2 Report
The manuscript describes putative beneficial effects of SKK on cultured hepatocytes. The current study provides data regarding the inhibition of FFA-induced cytotoxicity and intracellular lipid accumulation by SKK. Points to address:
- My major concern is the low number of experiments. In Figure Legend is indicated n=3 (replicates) and only 2 in gene expression experiments. It is not clear for me if this number is referred to independent experiments or replicates in the same experiment. Anyway, from my point of view, this “n” is not enough for supporting the conclusions.
- Figure legends text should be modified to provide the suitable methodological information. Figure legends should not explain the results.
- It is essential to discuss the transfer of the knowledge resulting from this in vitro study to the therapeutic benefits of SKK. Are SKK concentrations similar to those used in medical treatment? FFA dose is very high (2 mM). Taking into account that linoleic acid and palmitic acid are more abundant in plasma than oleic acid, were other combinations/proportions of FFA used?
- Did the authors measure any marker of inflammation? (protein in supernatant or at gene expression level). In this context, I think that to provide information about hepatic injuring enzymes is essential. In addition, was some oxidative stress marker evaluated?
- Some points of Methods are not clearly detailed. For example, after pre-treatment of HepG2 with SKK, was the medium removed and then added FFA? Was FFA incubated with albumin previously to their addition to cells?
- It would have been useful to corroborate the cytotoxicity results (found by Alamar blue) by other reliable techniques, such as flow cytometry, and to evaluate apoptosis
- The authors should evaluate ORO staining by an image program to corroborate the results found by absorbance quantification.
- The effect of SKK(in the absence of FFA) on intracellular lipid accumulation and gene expression should be indicated.
- Do the authors have any explanation for the observation that higher doses of SKK had a lower effect on gene expression (except for CPT1A)? CPT1A reduction by SKK should be further discussed, because the impairment of FFA oxidation is not a beneficial effect.
- Did the authors evaluate the expression of other important genes related to FFA/TG metabolism? I think that some additional genes should be evaluated, such as CD36, PPAR, DGAT, FABP1, PLIN, lipases, etc.
- Some comment should be included about the fact that intracellular FFA could yield to increase the levels of ceramide, a lipotoxic and inflammatory mediator.
- Revise mistakes in writing (for example line 371)
Author Response
Reviewer 2
Query 1: My major concern is the low number of experiments. In Figure Legend is indicated n=3 (replicates) and only 2 in gene expression experiments. It is not clear for me if this number is referred to independent experiments or replicates in the same experiment. Anyway, from my point of view, this “n” is not enough for supporting the conclusions.
Response: We thank the reviewer for pointing out the error. We have now replaced ‘n=3’ with the following statement, ‘All the experiments were performed are thrice in triplicates’ (3 biological and 3 technical) throughout the manuscript.
Query 2: Figure legends text should be modified to provide the suitable methodological information. Figure legends should not explain the results.
Response: We thank the reviewer for the comment. As per your suggestion, we have modified the figure legends by omitting the results. They now only describe the methodological information.
Query 3: It is essential to discuss the transfer of the knowledge resulting from this in vitro study to the therapeutic benefits of SKK. Are SKK concentrations similar to those used in medical treatment? FFA dose is very high (2 mM). Taking into account that linoleic acid and palmitic acid are more abundant in plasma than oleic acid, were other combinations/proportions of FFA used?
Response: This is an extremely relevant point. However, the current study is a follow-up of our previously published manuscript revealing the therapeutic benefit of SKK in ameliorating hepatic injuries induced by carbon tetrachloride done in Wistar rats (Balkrishna et al., 2020). In the present study, we had applied a half-log scale dose of SKK for studying its efficacy. We do acknowledge that it is difficult to convert the results obtained from the SKK treatments to human equivalent doses.
In the current study, we evaluated the efficacy of SKK in ameliorating FFA induced steatosis with doses of fatty acids previously reported by Fan et al., 2009, and Gómez-Lechón et al., 2007. The selction of oleic acid instead of linoleic acid was done as other fatty acids like linoleic acid and linolenic acid show a suppressive effect on the hepatic fatty acid de novo synthesis and fatty acid oxidation pathways (Kohjima et al., 2009). Based on these studies we selected the dose of 2 mM FFA combination of oleic acid and palmitic acid.
References:
- Balkrishna et al., “Polyherbal Medicine Divya Sarva-Kalp-Kwath Ameliorates Persistent Carbon Tetrachloride Induced Biochemical and Pathological Liver Impairments in Wistar Rats and in HepG2 Cells,” Front. Pharmacol., vol. 11, Mar. 2020, doi: 10.3389/fphar.2020.00288.
- Fan, Y. Chen, W. Bei, L. Wang, B. Chen, and J. Guo, “In Vitro Screening for Antihepatic Steatosis Active Components within Coptidis Rhizoma Alkaloids Extract Using Liver Cell Extraction with HPLC Analysis and a Free Fatty Acid-Induced Hepatic Steatosis HepG2 Cell Assay,” Evidence-Based Complement. Altern. Med., vol. 2013, pp. 1–9, 2013, doi: 10.1155/2013/459390.
- J. Gómez-Lechón, M. T. Donato, A. Martínez-Romero, N. Jiménez, J. V. Castell, and J.-E. O’Connor, “A human hepatocellular in vitro model to investigate steatosis,” Chem. Biol. Interact., vol. 165, no. 2, pp. 106–116, Jan. 2007, doi: 10.1016/j.cbi.2006.11.004.
- Kohjima et al., “The effects of unsaturated fatty acids on lipid metabolism in HepG2 cells,” Vitr. Cell. Dev. Biol. - Anim., vol. 45, no. 1–2, pp. 6–9, Feb. 2009, doi: 10.1007/s11626-008-9144-7.
Query 4: Did the authors measure any marker of inflammation? (protein in supernatant or at gene expression level). In this context, I think that to provide information about hepatic injuring enzymes is essential. In addition, was some oxidative stress marker evaluated?
Response: We again thank the reviewer for the excellent suggestion. We have now performed and added AST level analysis obtained from the FFA and SKK treated HepG2 cells (Figure 4A). The results showed that at the highest tested concentration of SKK (30 µg/mL) significant reduction of AST release from the HepG2 cells was observed.
We thank the reviewer for the comment. However we did not perform oxidative stress analysis in the present manuscript. In our earlier study, we have shown the an amelioration of oxidative stress induced by CCl4 in HepG2 cells (Balkrishna et al., 2020). We will include your suggestion in our future work related to steatosis.
Reference:
- Balkrishna et al., “Polyherbal Medicine Divya Sarva-Kalp-Kwath Ameliorates Persistent Carbon Tetrachloride Induced Biochemical and Pathological Liver Impairments in Wistar Rats and in HepG2 Cells,” Front. Pharmacol., vol. 11, Mar. 2020, doi: 10.3389/fphar.2020.00288.
Query 5: Some points of Methods are not clearly detailed. For example, after pre-treatment of HepG2 with SKK, was the medium removed and then added FFA? Was FFA incubated with albumin previously to their addition to cells?
Response: We apologise for the confusion. During the exposure, HepG2 cells were pre-treated for 24 h with SKK. Next day, the media was removed and co-treatment with FFA (2mM) and SKK (0-30 µg/ml) was given to the cells. We would also like to add that for preparation of media used for steatosis induction, we pre-incubated FFA in DMEM media containing 1% BSA for 18 hrs with constant shaking at 37 °C. This media was filtered through a 0.22- micron filter before exposure to the cells.
Query 6: It would have been useful to corroborate the cytotoxicity results (found by Alamar blue) by other reliable techniques, such as flow cytometry, and to evaluate apoptosis.
Response: We again thank the reviewer for his suggestions. However, we would like to state that performance of cytotoxicity evaluation using Alamar blue gave us the information regarding the cell viability as well as the metabolic functioning of HepG2 cells. So, additional analysis would only addon to the gathered data and not provide any new information. But we appreciate the recommendation and will be including the same in our further works.
Query 7. The authors should evaluate ORO staining by an image program to corroborate the results found by absorbance quantification.
Response: We appreciate the suggestion from the reviewer. However, microscopic quantification of ORO stained HepG2 using image processing programs is difficult due to the nature of HepG2 to grow in clusters. Proper tracing of cells using image software is not often accurate and the chances of false negatives are high in such state. So, quantification was done using multi-mode-plate reader for the whole cell population analysis following solubilization of the intracellular ORO stain using 2-Propanol (IPA).
Query 8. The effect of SKK (in the absence of FFA) on intracellular lipid accumulation and gene expression should be indicated.
Response: We thank the reviewer for the suggestion. We had actually performed a lipid accumulation assay in the HepG2 cells treated with SKK alone upto 48 h, using the HCS neutral lipid dye (ThermoFischer H34476). We did not observe any change in the intracellular level of lipid accumulation. Result from the study attached below-
Figure: HepG2 cells exposed to SKK alone for 24 and 48 h. Analysis was performed using HCS neutral lipid red fluorescent dye. Results displayed as mean±S.D. One-way ANOVA with Tukey’s multiple comparisons was performed. No statistically significant difference between control and SKK treatment was observed.
Query 9. Do the authors have any explanation for the observation that higher doses of SKK had a lower effect on gene expression (except for CPT1A)? CPT1A reduction by SKK should be further discussed, because the impairment of FFA oxidation is not a beneficial effect.
Response: We appreciate the keen observation of the reviewer. However, higher dose of SKK did not induce lower effect on gene expression. All the gene expression were observed similar to the normal control, which indicate that SKK ameliorated the steatosis responses at molecular level even in the presence of high FFA content. Regarding the CPT1A, we did see a down regulation of the gene following treatment with SKK. This is indeed a good indication as SKK did not reduce FFA metabolism below normal levels, that may lead to cellular energy starvation. Furthermore, it is well known that prolonged enhancement of CPT1A activity can lead to increased mitochondrial energy metabolism leading to heightened reactive oxygen species generation causing cellular damages and apoptosis (Brown et al., 2018).
Reference
Brown ZJ, Fu Q, Ma C, Kruhlak M, Zhang H, Luo J, Heinrich B, Yu SJ, Zhang Q, Wilson A, Shi ZD, Swenson R, Greten TF. Carnitine palmitoyltransferase gene upregulation by linoleic acid induces CD4+ T cell apoptosis promoting HCC development. Cell Death Dis. 2018 May 23;9(6):620. doi: 10.1038/s41419-018-0687-6. PMID: 29795111; PMCID: PMC5966464.
Query 10: Did the authors evaluate the expression of other important genes related to FFA/TG metabolism? I think that some additional genes should be evaluated, such as CD36, PPAR, DGAT, FABP1, PLIN, lipases, etc.
Response: We are grateful for this suggestion. The genes we selected form the basic group of genes affected by onset of steatosis. Therefore, these genes were used would exemplifying the efficacy of SKK in ameliorating FFA induced steatosis. Evaluation of the same set of gene was also done previously by Park et al, 2019 to showcase efficiency of a herbal drug against FFA induced steatosis in HepG2. We will definitely include the mentioned additional set of gene in our future research work on steatosis.
Reference
- Park, J.-H. Yoo, Y.-S. Lee, and H.-J. Lee, “Lonicera caerulea Extract Attenuates Non-Alcoholic Fatty Liver Disease in Free Fatty Acid-Induced HepG2 Hepatocytes and in High Fat Diet-Fed Mice,” Nutrients, vol. 11, no. 3, p. 494, Feb. 2019, doi: 10.3390/nu11030494.
Query 11: Some comment should be included about the fact that intracellular FFA could yield to increase the levels of ceramide, a lipotoxic and inflammatory mediator.
Response: This is an excellent suggestion from the reviewer. We have included the following information in manuscript-
Page 2 line 48: Introduction
Free fatty acids in diet also increase levels of ceramides which are known to be involved in several pathways linked to inflammation, apoptosis, insulin resistance and oxidative stress, and progression of steatosis.
Query 12: Revise mistakes in writing (for example line 371)
Response: We are grateful to the reviewer for the thorough reading of manuscript. We would like to add that the manuscript has been re-evaluated for writing errors and corrected for the same.

Reviewer 3 Report
In this manuscript, Balkrishna et al., studied the role Tri-herbal Decoction Divya Sarva-Kalp-Kwath (SKK) in Regulation of Fatty Acid-Induced Steatosis in cultured human heptoma HepG2 cell line, and rat primary hepatocytes. The work is interesting, but the manuscript is not well written.
Comments:
What’s the rational to use Human HepG2 and rat hepatocytes? There are human hepatocytes available.
Title: change “Human Liver (HepG2) Cells” to Human HepG2 cells.
2.1. HPTLC fingerprinting of Divya-Sarva-Kalp-Kwath (SKK). Too much details on technical side, you need to give the levels of phonolic compounds (gallic acid, caffeic acid, quercetin etc. in the SKK (ug/mg).
Fig 2. SKK tested concentration ued was up to 1000 µg/ml, what’s the solvent?
SKK (30 μg/ml) co-treatment for 48 hr showed a dose-dependent amelioration of FFA-induced cell-death”. It promoted cell growth by approximately 20%, please explain the mechanisms.
Lines 123-124, “Overall results indicated a protective role for SKK against FFA-induced toxicity without inducing any additional side-effects”. What do you mean side-effects here?
Line 252, “the human hepatocytes (HepG2)”, this is not correct expression.
Figure 5. Gene expression analysis in HepG2 cells treated with free fatty acids and SKK. Do you have Western blot results to support these gene expression data?
Author Response
Reviewer 3
Query 1: What’s the rational to use Human HepG2 and rat hepatocytes? There are human hepatocytes available.
Response: We thank the reviewer for the query. The present study was performed as a preliminary screen of SKK efficacy in mitigating free fatty acid (FFA) induced steatosis. The study is also a follow-up of our previously published research on the hepatoprotective effect of SKK in reducing CCl4 induced toxicity (Balkrishna et al., 2020). The current research work has been performed using the study model for NAFLD established by other eminent researchers such as, Stellavato et al. 2018, Go et al. 2017, Fan et al. 2013, and Park, et al. 2017. As a future plan, we will follow the suggestion provided by the reviewer and apply SKK in studying FFA induced steatosis in human primary hepatocytes.
References:
Balkrishna et al., “Polyherbal Medicine Divya Sarva-Kalp-Kwath Ameliorates Persistent Carbon Tetrachloride Induced Biochemical and Pathological Liver Impairments in Wistar Rats and in HepG2 Cells,” Front. Pharmacol., vol. 11, Mar. 2020, doi: 10.3389/fphar.2020.00288.
Stellavato, A., Pirozzi, A.V.A., de Novellis, F. et al. In vitro assessment of nutraceutical compounds and novel nutraceutical formulations in a liver-steatosis-based model. Lipids Health Dis 17, 24 (2018). doi: 10.1186/s12944-018-0663-2
Go, J. A. Ryuk, J. T. Hwang, and B. S. Ko, Effects of three different formulae of Gamisoyosan on lipid accumulation induced by oleic acid in HepG2 cells, Integr. Med. Res. 6, 395–403 (2017). doi: 10.1016/j.imr.2017.08.004.
Fan, Y. Chen, W. Bei, L. Wang, B. Chen, and J. Guo, “In Vitro Screening for Antihepatic Steatosis Active Components within Coptidis Rhizoma Alkaloids Extract Using Liver Cell Extraction with HPLC Analysis and a Free Fatty Acid-Induced Hepatic Steatosis HepG2 Cell Assay,” Evidence-Based Complement. Altern. Med., vol. 2013, pp. 1–9, 2013, doi: 10.1155/2013/459390.
Park, Y., Sung, J., Yang, J. et al. Inhibitory effect of esculetin on free-fatty-acid-induced lipid accumulation in human HepG2 cells through activation of AMP-activated protein kinase. Food Sci Biotechnol 26, 263–269 (2017). https://doi.org/10.1007/s10068-017-0035-0
Query 2: Title: change “Human Liver (HepG2) Cells” to Human HepG2 cells.
Response: We thank the reviewer for suggesting the change. We have now changed the title of the manuscript to-
“Tri-herbal Medicine Divya Sarva-Kalp-Kwath (Levogrit) Regulates Fatty Acid-Induced Steatosis in Human HepG2 Cells Through Inhibition of Intracellular Triglycerides and Extracellular Glycerol Levels”
Query 3: HPTLC fingerprinting of Divya-Sarva-Kalp-Kwath (SKK). Too much details on technical side, you need to give the levels of phenolic compounds (gallic acid, caffeic acid, quercetin etc. in the SKK (ug/mg).
Response. We thank the reviewer for the comment. We have now added the total amount of polyphenols present in SKK analysed using Folin-Ciocalteu method. We have now included in the manuscript-
Page 14, Line 393: Material and Methods
Total phenolic content of SKK was also analyzed via Folin Ciocalteu method using gallic acid as a standard [48]
Page 3, Line 116: Results
The amount of total polyphenol present in SKK, obtained using the Folin- Ciocalteu method was 4.22 % w/w. Based upon HPTLC fingerprinting and phenolic content analysis, we re-affirmed the chemical constituents of SKK.
Reference.
Blainski, A.; Lopes, G.C.; de Mello, J.C. Application and analysis of the folin ciocalteu method for the determination of the total phenolic content from Limonium brasiliense L. Molecules 2013, 18, 6852-6865, doi:10.3390/molecules18066852.
Query 4: SKK tested concentration used was up to 1000 µg/ml, what’s the solvent?
Response: We thank the reviewer for the query. For the study, SKK stock concentration was prepared by suspending 30 mg/ml of dry SKK powder in sterile PBS containing 1% BSA. Working concentration of SKK were prepared in DMEM media containing 1% BSA imideately before exposure to the cells.
Query 5: SKK (30 μg/ml) co-treatment for 48 hr showed a dose-dependent amelioration of FFA-induced cell-death”. It promoted cell growth by approximately 20%, please explain the mechanisms.
Response: We are grateful for the keen observation of the reviewer. However, HepG2 cells treated with the 2:1 combination of Oleic acid (O) and Palmitic acid (P), and 30 µg/ml of SKK showed an increase in cell viability up to 113.4 ± 12.07% (SEM), which is well within the acceptable range (see Figure 2B).
Query 6: Lines 123-124, “Overall results indicated a protective role for SKK against FFA-induced toxicity without inducing any additional side-effects”. What do you mean side-effects here?
Response: We appreciate the query made by the reviewer. We have now further elaborated the statement by modifying it-
Page 4, line 139: Results
These preliminary results indicated a protective role for SKK against FFA-induced toxicity without inducing any other observable side-effects like decrease in cell viability or metabolic impairment.
Query 7: Line 252, “the human hepatocytes (HepG2)”, this is not correct expression.
Response: We appreciate the suggestion of the reviewer. We have now replaced it with human HepG2 cells or simply HepG2 cells through out the manuscript.
Query 8: Figure 5. Gene expression analysis in HepG2 cells treated with free fatty acids and SKK. Do you have Western blot results to support these gene expression data?
Response: We appreciate the remark of the reviewer. We would like to state that the change in expression of genes used for this study are mentioned in some previous studies by Park et al., 2019 (as shown in figure below) wherein the protein level of lipogenic factors, SREBP-1c, C/EBPα, FAS were correlated with their respective expression profiles. As, we got similar findings in our report we were able to co-relate our findings to the same and did not proceed with Western blot analysis.
References
Park, J.-H. Yoo, Y.-S. Lee, and H.-J. Lee, “Lonicera caerulea Extract Attenuates Non-Alcoholic Fatty Liver Disease in Free Fatty Acid-Induced HepG2 Hepatocytes and in High Fat Diet-Fed Mice,” Nutrients, vol. 11, no. 3, p. 494, Feb. 2019, doi: 10.3390/nu11030494.

Round 2
Reviewer 1 Report
The authors took into consideration all my previous concerns. The current version of the manuscript has now been considerably improved and meets all quality criteria for publication.
Author Response
We are thankful to the reviewer for all the previous inputs and remarks that led to a substantial addon in the scientific and literary structure of our manuscript. We are grateful for the reviewer for thoroughly going through the manuscript and pointing out the clarifications that needed to be added in our manuscript. We have amended the manuscript as per your kind suggestions.
Reviewer 2 Report
My major concern is still the low number of experiments, because 3 experiments are not enough for supporting the conclusions. This is a clear limitation of the study. In this context, it is confusing that the first version of the manuscript indicates n=2 in Figure 5, but, now, it is stated that the experiments were performed thrice.
Some of my recommendations have been partially addressed. Some of the answers should be included in the text, such as the effect of SKK (in the absence of FFA) in gene expression and ORO stainning. In addition, several points should be further discussed in the manuscript, for example: why other combinations of FFA (more similar to those found physiologically) were discarded, or the putative negative consequences of CPT1A down-regulation by SKK (in the presence of high FFA concentrations).
Regarding query 11, I meant that FFA could mediate the effects found in the study through induction of intracellular ceramide synthesis. Some comment should be included in Discussion.
Author Response
Query 1: My major concern is still the low number of experiments, because 3 experiments are not enough for supporting the conclusions. This is a clear limitation of the study. In this context, it is confusing that the first version of the manuscript indicates n=2 in Figure 5, but now, it is stated that the experiments were performed thrice.
Response: We thank the reviewer and humbly disagree since the experiments were performed thrice in triplicate in case of all the experiments except in the case of mRNA expression where the experiments were repeated twice in triplicate. In this context, the experiments consisted of three technical replicated and twice/thrice in terms of biological replicate. This means that each data point represents a total of 9 individual replicates. We observed statistically significant differences (p <0.05) between normal control and free fatty acid alone treated cells using one-way ANOVA with Dunnett's multiple comparisons post-hoc test. In the context of Figure 5, we thank the reviewer for comment, we have now elaborated it as ‘thrice in duplicate’ meaning the experiments represent three biological replicates and two technical replicates.
Query 2: Some of my recommendations have been partially addressed. Some of the answers should be included in the text, such as the effect of SKK (in the absence of FFA) in gene expression and ORO staining.
Response: We thank the reviewer again for the excellent comment. Based on the previous and current suggestions, we have included the following information in the revised manuscript-
Page 5, Line 15:1 Results
Qualitative analysis of lipid accumulation in the HepG2 using ORO dye showed that cells treated with varying doses of SKK up to the highest therapeutic dose did not show any change in intracellular lipid accumulation compared to normal control (Figure 3B i, 3B ii). However, cells treated with 2 mM FFA showed a significant increase in the intracellular lipid accumulation indicating the onset of steatosis (as seen in Figures 3B iii). Pre-treatment of the HepG2 cells with varying concentrations of SKK (3, 10, and 30 µg/ml) considerably reduced the FFA-induced accumulation of intracellular lipids in the cells (Figures 3B iv, 3B v, 3B vi).
Regarding gene expression studies of SKK treatment alone in HepG2 cells, we did not proceed with the assay, as cell viability and ORO assays indicated the herbal formulation to be benign on its own by not inducing loss of cell viability and accumulation of intracellular lipids (see Figures 2A and 23B ii). We have now added the following in the discussion part-
Page 13, Line 358: Discussion
We did not investigate the role of SKK alone (without FFA) in stimulating steatosis associated genes since SKK appeared benign in terms of inducing any loss of cell viability or intracellular accumulation of lipids in the HepG2 cells.
Query 3: In addition, several points should be further discussed in the manuscript, for example: why other combinations of FFA (more similar to those found physiologically) were discarded, or the putative negative consequences of CPT1A down-regulation by SKK (in the presence of high FFA concentrations).
Response: Thank you for the remark. As we have earlier mentioned in our round 1 review, the selection of oleic acid instead of linoleic acid was done since the latter suppresses the effect on the hepatic fatty acid through inhibition of de novo synthesis and fatty acid oxidation pathways (Kohjima et al., 2009). Furthermore, high-oleic acid and high-palmitic acids obtained as a food source has been shown to induce steatosis in hepatocytes of different organisms (Joshi-Barve et al. 2007, Tang et al. 2011, Niklas et al. 2012, Pan et al. 2011, Gómez-Lechón et al. 2007, Mantzaris et al. 2011, Ricchi et al. 2009). Therefore, we have added the following lines in the revised manuscript-
Page 2, Line 53: Introduction
Palmitic acid and monounsaturated oleic acid represents the two most abundant FFA present in high-fat diets [6]. Intake of high levels of oleic acid and palmitic acids from food sources has been shown to induce steatosis in hepatocytes of different organisms [7-14].
Page 11, Line 291. Discussion
Intracellular increase in the triglyceride and extracellular lipid levels decrease in the glycerol level are the hallmarks of steatosis ailment [41]. Studies on steatosis seldom emphasize the diet-related increase in hepatic lipids and remain largely focused on the transcriptional segment of fatty acid synthesis enzymes. In our present study, we have selected and utilized 2 mM concentration of FFA for induction of steatosis as other extracellular FFA concentrations did not induce a sizable fat accumulation in the HepG2 cells, under present experimental conditions. The observations are in line with the previous works of Lechón et al. and Ricchi et al. [11,13]. Therefore, this in vitro model might closely resemble the liver ailment conditions observed clinically in patients.
Regarding the up-/ down-regulation of the CPT-1a gene in the FFA and SKK treated HepG2 cells, we have added the following-
Page 12, Line 332. Discussion
An increase in fatty acids is responsible for induction of CPT1A levels in the liver [56,57]. In the present study, CPT1A expression increased in HepG2 cells stimulated with 2 mM FFA. SKK treatment brought down this upregulated CPT1A levels to the normal level. This apparent normalization of CPT1A levels post SKK treatment points towards a chain of events leading to the attainment of homeostasis by the cells [58,59]. Prolonged enhancement of CPT-1A activity can lead to increased mitochondrial energy metabolism leading to heightened reactive oxygen species generation and causing cellular damage and apoptosis [60]. Whereas, a decrease in CPT1A expression can lead to reduced β‐oxidation in liver and could contribute to fatty acid accumulation and inflammation in hepatocytes [61].
Reference
- Kohjima et al., “The effects of unsaturated fatty acids on lipid metabolism in HepG2 cells,” Vitr. Cell. Dev. Biol. - Anim., vol. 45, no. 1–2, pp. 6–9, 2009, doi: 10.1007/s11626-008-9144-7.
- Joshi-Barve S, et al., Palmitic acid induces production of proinflammatory cytokine interleukin-8 from hepatocytes. Hepatology 46: 823-830, 2007.
- Tang Y, et al., MA X: Interleukin-17 exacerbates hepatic steatosis and inflammation in non-alcoholic fatty liver disease. Clin Exp Immunol 166: 281-290, 2011.
- Niklas J, et al., Central energy metabolism remains robust in acute steatotic hepatocytes challenged by a high free fatty acid load. BMB Rep 45: 396-401, 2012.
- Pan Z, et al. Effects of palmitic acid on lipid metabolism homeostasis and apoptosis in goose primary hepatocytes. Mol Cell Biochem 350: 39-46, 2011.
- Gómez-Lechón MJ, et al. A human hepatocellular in vitro model to investigate steatosis. Chem Biol Interact 165: 106-116, 2007.
- Mantzaris MD, et al. Interruption of triacylglycerol synthesis in the endoplasmic reticulum is the initiating event for saturated fatty acid-induced lipotoxicity in liver cells. FEBS J 278: 519-530, 2011.
- Ricchi M, et al. Differential effect of oleic and palmitic acid on lipid accumulation and apoptosis in cultured hepatocytes. J Gastroenterol Hepatol 24: 830-840, 2009
Query 4: Regarding query 11, I meant that FFA could mediate the effects found in the study through induction of intracellular ceramide synthesis. Some comment must be included in the Discussion.
Response: We again thank the reviewer for his suggestion. We have now added the following sentence regarding the relation of FFA and ceramides in our manuscript-
Page 11, Line 286. Discussion
High levels of fatty acids obtained from the diet or those produced after de novo lipogenesis increase the hepatic influx of fatty acids and stimulate ceramide synthesis. The de novo ceramide synthesis that occurs using serine and palmitoyl CoA in the endoplasmic reticulum, is upregulated in presence of an excess of saturated fatty acids. Thus, an increase in ceramide levels has been known to promote triglyceride synthesis, induce hepatic lipid accumulation, lipotoxicity, and provoke apoptosis [39,40].

Reviewer 3 Report
I can recommend to accept this revised version for publication.
Author Response
We highly appreciate that the reviewer thinks that the manuscript is ready for publication. The noteworthy inclusions and clarifications pointed out by the reviewer have contributed substantially to the quality of our manuscript. We are also grateful for the literary amendments suggested by the reviewer. We again like to acknowledge the reviewer for going through the manuscript and providing valuable inputs for the betterment of our work.
This manuscript is a resubmission of an earlier submission. The following is a list of the peer review reports and author responses from that submission.